# Targeting senescent cells enhances adipogenesis and metabolic function in old age

Ming Xu[†], Allyson K Palmer[†], Husheng Ding, Megan M Weivoda, Tamar Pirtskhalava, Thomas A White, Anna Sepe, Kurt O Johnson, Michael B Stout, Nino Giorgadze, Michael D Jensen, Nathan K LeBrasseur, Tamar Tchkonia, James L Kirkland*

Robert and Arlene Kogod Center on Aging, Mayo Clinic, Rochester, United States

**Abstract** Senescent cells accumulate in fat with aging. We previously found genetic clearance of senescent cells from progeroid INK-ATTAC mice prevents lipodystrophy. Here we show that primary human senescent fat progenitors secrete activin A and directly inhibit adipogenesis in non-senescent progenitors. Blocking activin A partially restored lipid accumulation and expression of key adipogenic markers in differentiating progenitors exposed to senescent cells. Mouse fat tissue activin A increased with aging. Clearing senescent cells from 18-month-old naturally-aged INK-ATTAC mice reduced circulating activin A, blunted fat loss, and enhanced adipogenic transcription factor expression within 3 weeks. JAK inhibitor suppressed senescent cell activin A production and blunted senescent cell-mediated inhibition of adipogenesis. Eight weeks-treatment with ruxolitinib, an FDA-approved JAK1/2 inhibitor, reduced circulating activin A, preserved fat mass, reduced lipotoxicity, and increased insulin sensitivity in 22-month-old mice. Our study indicates targeting senescent cells or their products may alleviate age-related dysfunction of progenitors, adipose tissue, and metabolism.

*For correspondence: Kirkland. James@mayo.edu

[†]These authors contributed equally to this work

## Introduction

A major function of adipose tissue is to store potentially cytotoxic lipids, including fatty acids (FAs), as less reactive neutral triglycerides (TG) within fat droplets (*Listenberger et al., 2003*). Lipid storage by adipose tissue appears to constitute a defense against lipotoxicity and metabolic disease (*Wang et al., 2008*; *Unger and Scherer, 2010*; *Gustafson et al., 2015*; *Tchkonia et al., 2010*). Fat cells turn over throughout life, with generation of new fat cells through differentiation of fat progenitors (also known as preadipocytes or adipose-derived stem cells) (*Tchkonia et al., 2013*; *Spalding et al., 2008*; *Tchoukalova et al., 2010*). Adipogenesis is orchestrated by a transcription factor cascade involving the two key regulators, peroxisome proliferator-activated receptor-γ (PPARγ) and CCAAT/enhancer binding protein-α (C/EBPα) (*Wu et al., 1999*; *Lin and Lane, 1994*) and their downstream targets, including fatty acid binding protein 4 (*FABP4*) and perilipin (*PLIN1*) (*Bernlohr et al., 1997*; *Sun, 2013*). Compromised adipogenic capacity can contribute to impaired ability of adipose tissue to store lipids, leading to FA spillover and ectopic lipid accumulation in liver and other sites, insulin resistance, and lipotoxicity (*Garbarino and Sturley, 2009*; *Slawik and Vidal-Puig, 2006*; *Tchkonia et al., 2006*; *Guo et al., 2007*). By late middle age, capacity for adipogenesis, *PPARγ* and *C/EBPα* expression, adipose tissue mass, and metabolic function begin to decline in experimental animals and humans (*Tchkonia et al., 2010*; *Slawik and Vidal-Puig, 2006*; *Fink et al., 1983*; *Tchkonia et al., 2013*; *Cowie et al., 2006*; *North and Sinclair, 2012*; *Palmer et al., 2015*; *Cartwright et al., 2007*; *Raguso et al., 2006*; *Kuk et al., 2009*; *Cartwright et al., 2010*;

**eLife digest** The likelihood of developing metabolic diseases such as diabetes increases with age. This is, in part, because the cells within fat and other tissues become less sensitive to the hormone insulin as people and other animals get older. Also, the stem cells that give rise to new, insulin-responsive fat cells become dysfunctional with increasing age. This is related to the accumulation of "senescent" cells, which, unlike normal fat cell progenitors, release molecules that are toxic to nearby and distant cells.

Xu, Palmer et al. have now investigated if senescent cells interfere with the activity of stem cells from human fat tissue, and if getting rid of these senescent cells might restore the normal activity and insulin responsiveness of aged fat tissue. The experiments revealed that human senescent fat cell progenitors release a protein called activin A, which impedes the normal function of stem cells and fat tissue. Additionally, older mice had higher levels of activin A in both their blood and fat tissue than young mice.

Xu, Palmer et al. then analyzed older mice that had been engineered to have senescent fat cells that could be triggered to essentially kill themselves when the mice were treated with a drug. Eliminating the senescent cells from these mice led to lower levels of activin A and more fat tissue (due to improved stem cell capacity to become fully functional fat cells) that expressed genes required for insulin responsiveness. This showed that senescent cells are a cause of age-related fat tissue loss and metabolic disease in older mice.

Next, Xu, Palmer et al. treated older mice with drugs called JAK inhibitors, which they found reduce the production of activin A by senescent cells isolated from fat tissue. After two months of treatment, the levels of activin A in the blood and in fat tissue were indeed reduced. The fat tissue in treated mice also showed fewer features associated with the development of diabetes than the fat tissue of untreated mice. As such, these results paralleled those after selectively eliminating the senescent cells.

Together these findings suggest that JAK inhibitors or drugs (called senolytics) that selectively eliminate senescent cells may have clinical benefits in treating age-related conditions such as diabetes and stem cell dysfunction.

*Tchkonia et al., 2007*; *Karagiannides et al., 2001*; *Kirkland et al., 1990*). This age-related lipodystrophy likely contributes to the pathogenesis of metabolic dysfunction at older ages (*Gustafson et al., 2015*; *Tchkonia et al., 2010*; *Tchkonia et al., 2006*; *Guo et al., 2007*; *Kuk et al., 2009*).

We hypothesize that cellular senescence could contribute to impaired adipogenesis and age-related lipodystrophy (*Tchkonia et al., 2010*). Cellular senescence refers to an essentially irreversible arrest of cell proliferation (*Hayflickl and Moorhead, 1961*). It can be induced by a variety of stresses, including DNA damage, telomere shortening, radiation, chemotherapeutics, and reactive metabolites (*Tchkonia et al., 2013*; *Campisi and d'Adda di Fagagna, 2007*). Senescent cells accumulate in adipose tissue with aging across a number of mammalian species (*Tchkonia et al., 2010*; *Xu et al., 2015*; *Stout et al., 2014*) and secrete an array of cytokines, chemokines, proteases, and growth factors—the senescence-associated secretory phenotype (SASP) (*Coppé et al., 2008*; *Coppé et al., 2010*). Cultures of progenitors isolated from adipose depots of older animals or humans contain senescent cells and exhibit impaired adipogenic capacity, with reduced lipid accumulation and C/EBPα and PPARγ expression after exposure to differentiation-inducing stimuli (*Tchkonia et al., 2010*; *Tchkonia et al., 2007*; *Park et al., 2005*; *Mitterberger et al., 2014*). Senescent cells appear to be able to spread inflammatory activation and perhaps even senescence to nearby non-senescent cells (*Xu et al., 2015*; *Acosta et al., 2013*; *Nelson et al., 2012*). In previous work, we used a genetically modified INK-ATTAC (*Cdkn2a /p16$^{Ink4a}$* promoter driven apoptosis through targeted activation of caspase) mouse model to selectively eliminate *Cdkn2a (p16$^{Ink4a}$)* positive senescent cells through apoptosis by the administration of AP20187, a drug that induces dimerization of a membrane-bound myristoylated FK506 binding protein fused with caspase 8 (FKBP–Casp8) (*Baker et al., 2011*). We showed that clearance of senescent cells can delay age-related

phenotypes including lordokyphosis and cataract formation, and can actually reverse age-related fat loss in progeroid *BubR1*$^{H/H}$ animals (*Baker et al., 2011*), implicating senescent cells as a driver of age-related phenotypes. Furthermore, interleukin-6 (IL6) (*Gustafson and Smith, 2006*; *Okada et al., 2012*), tumor necrosis factor α (TNFα) (*Tchkonia et al., 2007*; *Gustafson and Smith, 2006*; *Okada et al., 2012*), and interferon γ (IFNγ) (*McGillicuddy et al., 2009*) can inhibit adipogenesis in vitro. These factors are among the SASP components in senescent fat progenitors and other senescent cell types (*Tchkonia et al., 2013*; *Xu et al., 2015*; *Coppé et al., 2008*; *Coppé et al., 2010*). However, causal links between these paracrine factors and impaired adipogenesis related to cellular senescence have not been demonstrated. We recently reported that the JAK/STAT (Janus kinase/ signal transducer and activator of transcription) pathway plays a role in regulating the SASP (*Xu et al., 2015*). Therefore, we hypothesized that JAK inhibition might rescue impaired adipogenesis due to senescent cells and thus preserve fat mass and metabolic function in older individuals.

We report here that senescent fat progenitors impede differentiation of non-senescent progenitors, in part by secreting activin A, a member of the transforming growth factor superfamily, which can inhibit adipogenesis and interfere with stem cell and progenitor function (*Zaragosi et al., 2010*). Eliminating senescent cells from naturally-aged INK-ATTAC mice reduced activin A and increased adipose tissue *C/EBPα* and *PPARγ*. JAK pathway inhibition suppressed production of activin A by senescent fat progenitors and partially rescued adipogenic capacity both in vitro and in vivo. JAK inhibition in aged mice reduced lipotoxicity and increased insulin sensitivity. Our findings provide new insights into the mechanisms of age-related progenitor dysfunction, fat loss, and metabolic dysfunction, as well as potential therapeutic avenues for preventing or alleviating these common conditions.

## Results

### Senescent fat cell progenitors impede adipogenesis

To determine if senescent cells influence adipogenesis in adjacent non-senescent cells, we devised a co-culture system with non-senescent human primary fat progenitors as 'target' cells and either senescent or non-senescent human progenitors as 'source' cells. Primary cells were isolated from the stromal-vascular fraction of collagenase-digested subcutaneous fat from healthy human subjects undergoing surgery to donate a kidney. Cells were passaged 4–6 times under conditions to enrich for fat progenitors as opposed to endothelial cells or macrophages (*Tchkonia et al., 2013*). These cells were exposed to 10 Gy irradiation, which induced at least 70% of cells to become senescence-associated β-galactosidase (SABG) -positive within 20 days, as previously described (*Xu et al., 2015*). Target cells were distinguished from source cells by fluorescent labeling (CM-DiI), which does not independently affect adipogenesis. We differentiated the mixture of cells using an adipogenic differentiation medium (DM) for 15 days. Differentiation was assessed by examining lipid accumulation inside the cells. We considered a cell to be differentiated if it contained doubly refractile lipid droplets visible by low power phase contrast microscopy, a change that occurs in fat cell progenitors following DM exposure, but not in other cell types (*Karagiannides et al., 2006*). We found that senescent source cells were less differentiated than control non-senescent source cells (*Figure 1a*). When co-cultured with senescent source cells, only 20% of target progenitors accumulated lipid compared to more than 50% when co-cultured with non-senescent source cells (*Figure 1b*), indicating that senescent cells can directly impair lipid accumulation by nearby fat progenitors.

Next, we examined the nature of the factors responsible for impairing adipogenesis. Non-senescent progenitors were treated with DM in the presence of conditioned medium (CM) from cultures of senescent or non-senescent cells. Senescent progenitor CM reduced differentiated cell numbers in target non-senescent cells at all three time points tested (*Figure 2a*). PPARγ, C/EBPα, FABP4, and PLIN2 are normally up-regulated during adipogenesis (*Wu et al., 1999*; *Lin and Lane, 1994*; *Bernlohr et al., 1997*; *Sun, 2013*). Differentiation-dependent expression of these genes was blunted by CM from senescent cells compared to CM from blank culture flasks or control non-senescent cells (*Figure 2b*). Cellular senescence did not appear to be induced in the target cells by the 15 days of CM exposure, since *p16*$^{Ink4a}$ and *Cdkn1a (p21*$^{Cip}$) transcript levels were not increased (*Figure 2—figure supplement 1a*). Adipogenesis was not impaired when exposure to CM was limited to 24 hours

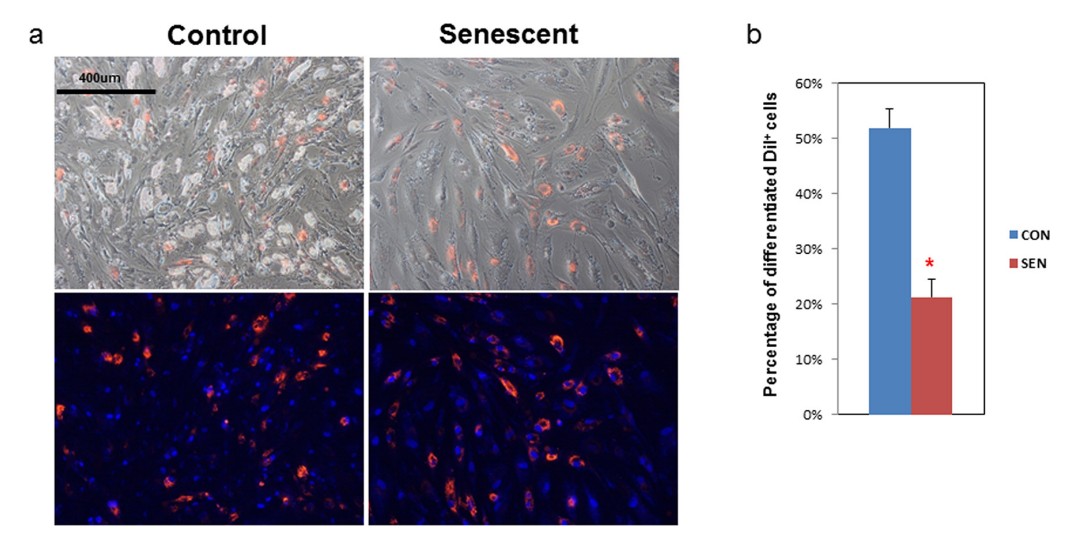

**Figure 1.** Adipogenesis in human fat progenitors is impeded by co-culture with senescent cells. Primary subcutaneous human fat progenitors were labelled with DiI and seeded into wells containing either control or radiation-induced senescent preadipocytes. (**a**) Photographs were taken 15 days after initiating differentiation. Representative images are shown. DiI-positive cells are red and DAPI staining is blue. (**b**) Number of differentiated DiI positive cells as a percentage of total DiI positive cells is expressed as mean ± s.e.m. *p<0.00001. Results were obtained using separate strains of fat progenitors harvested from 6 healthy human subjects during surgery to donate a kidney (N=6). Two-tailed Student's t tests were used to determine statistical significance.

The following source data is available for figure 1:

**Source data 1.** Adipogenesis in human fat progenitors is impeded by co-culture with senescent cells.

of pretreatment before exposure to DM (*Figure 2—figure supplement 1b*). This suggests that impaired adipogenesis due to senescent CM depends on continued presence of products secreted by senescent cells. CM from doxorubicin-induced senescent cells suppressed adipogenesis similarly to CM from irradiation-induced senescent cells (*Figure 2—figure supplement 1c*).

## Inhibition of activin A rescues impaired adipogenesis due to senescent CM

We next investigated which factors secreted by senescent cells impair adipogenesis. We found that CM from senescent cells inhibited adipogenesis even after freeze-thaw cycles (*Figure 2a,b*). Therefore, cell-cell contact or molecules with short half-lives, including many metabolites such as reactive oxygen species (ROS), do not appear to be the sole responsible factors. CM from senescent cells was separated into two fractions using molecular size filters with a cutoff at ~10 kd. The fraction larger than ~10kd impaired adipogenesis while the fraction smaller than ~10kd had no effect (*Figure 2—figure supplement 1d*). This led us to hypothesize SASP peptides or proteins might play a role in the inhibition of adipogenesis. Using either neutralizing antibodies or specific inhibitors, we inhibited candidate SASP factors in the CM, including IL6, TNFα, IFN γ, and activin A, which can be secreted by senescent cells and inhibit adipogenesis (*Tchkonia et al., 2007*; *Xu et al., 2015*; *Gustafson and Smith, 2006*; *Okada et al., 2012*; *McGillicuddy et al., 2009*; *Zaragosi et al., 2010*) (*Figure 2— figure supplement 1e*). Among the compounds screened, SB-431542, an activin A receptor inhibitor (*Inman et al., 2002*), substantially improved adipogenesis in progenitors exposed to CM from senescent cells, while only slightly increasing adipogenesis in control cells (*Figure 3a,b*). Due to the fact that SB-431542 also inhibits TGFβ signaling (*Inman et al., 2002*), to confirm further the role of activin A, we used activin A-specific neutralizing antibody and observed a similar enhancement of adipogenesis (*Figure 3c,d*). Together, these findings indicate that activin A plays a role in the impairment of adipogenesis by senescent cells.

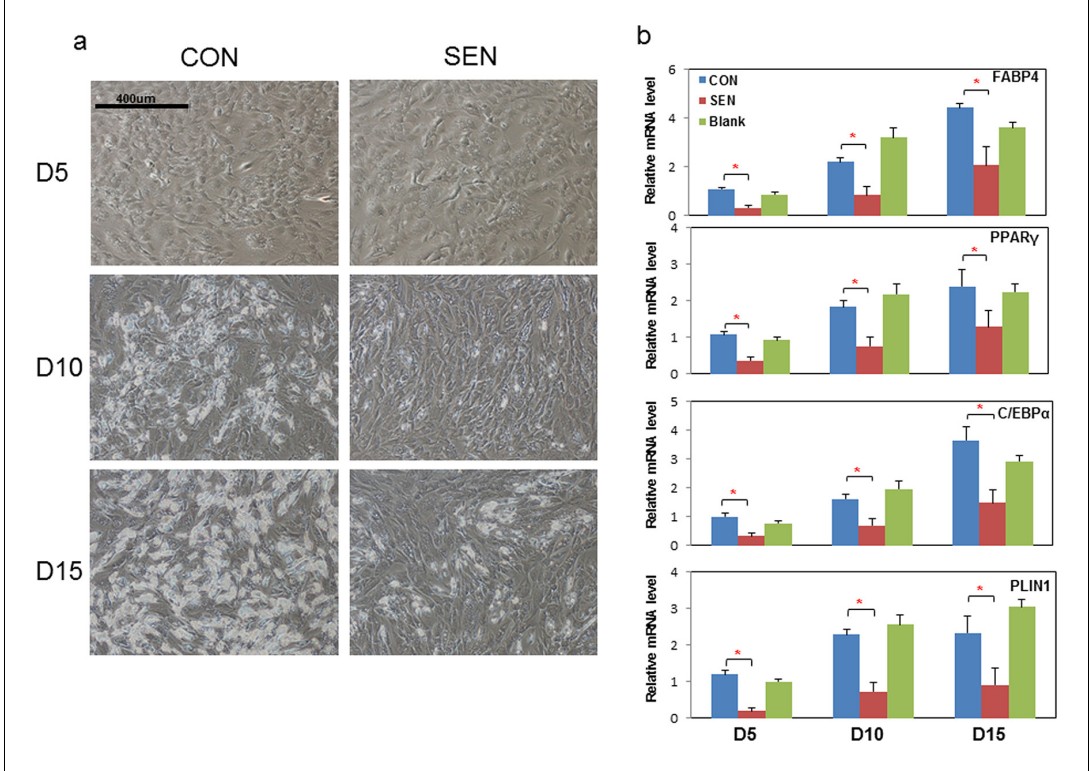

**Figure 2.** Conditioned medium from senescent cells impedes adipogenesis in human progenitors. Conditioned medium (CM) was collected from a flask with no cells present (Blank), control non-senescent (CON), and senescent (SEN) fat progenitor cultures. Pooled human progenitors from subcutaneous fat of 5 healthy subjects were treated with 50:50% CM:differentiation medium (DM) for 15 days. (a) Representative images are shown at day 5, 10, and 15 of exposure to CM + DM. (b) Gene expression was analyzed by real-time PCR at day 5, 10, and 15 of exposure to CM + DM. Results are shown as fold change relative to the CON group at day 5. Results were obtained using CM from 5 strains of human primary fat progenitors from different subjects and expressed as mean ± s.e.m. *p<0.05. Two-tailed Student's t tests were used to determine statistical significance.

The following source data and figure supplement are available for figure 2:

**Source data 1.** Conditioned medium from senescent cells impedes adipogenesis in human progenitors.

**Figure supplement 1.** Senescent cells impede adipogenesis in fat progenitors.

## Genetic clearance of senescent cells blunts fat loss and increases adipogenesis in 18-month-old mice

After 17–18 months-of-age, mice begin to lose fat mass. We previously found that senescent cells start to accumulate noticeably before 18 months-of-age in mouse fat tissue (*Stout et al., 2014*) and senescent cells play a role in age-related loss of subcutaneous fat in animals with progeria (*Baker et al., 2011*). However, it is still unknown whether senescent cell clearance has effect on age-related adipose phenotypes in naturally aged mice. To test this, we treated late middle-aged (18-month-old) INK-ATTAC[+/-] and wild-type (WT) littermates with two 3-day courses of AP20187, with 14 days between treatments, for 3 weeks (total 6 days of treatment) to activate the caspase-8 moiety in the *ATTAC* suicide gene product that is expressed only in $p16^{Ink4a}$ positive senescent cells. This allowed us to investigate the short-term response to senescent cell clearance, for example effects on adipogenic transcription factors, and to reduce effects of possible long-term compensatory responses. During the three-week treatment period, WT mice lost more fat than INK-ATTAC[+/-] mice (*Figure 4a*), while lean mass (*Figure 4b*) and total body weight (*Figure 4c*) were unaffected. Circulating activin A was reduced more than 30% compared to baseline in the INK-ATTAC[+/-] mice, while activin A increased by 10% in the WT group (*Figure 4e*). Activin A was also reduced in adipose tissue of the INK-ATTAC[+/-] mice (*Figure 4f*). Adipose tissue expression of C/EBPα and PPARγ was

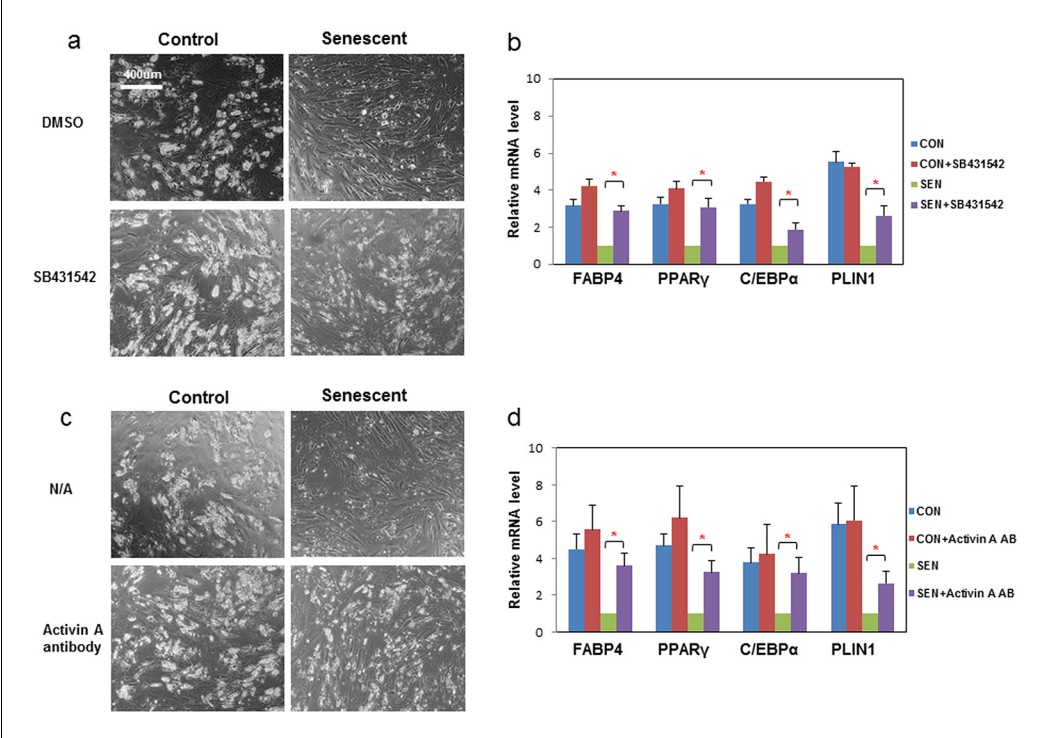

**Figure 3.** Inhibition of activin A alleviates the impairment of adipogenesis induced by senescent progenitors. CM was collected from control (CON) and senescent (SEN) fat progenitors. Pooled human progenitors were treated with a 50:50 mixture of CM:DM in the presence of DMSO or 5μM SB431542 (SB431542). (a) Representative images are shown of differentiated cells at day 15. (b) RNA was collected 7 days after differentiation and real-time PCR was performed. Pooled human progenitors were treated with a 50:50 mixture of CM:DM in the presence or absence of 1μg/ml activin A neutralizing antibody (Activin A AB). (c) Representative images are shown of differentiated cells at day 15. (d) RNA was collected 7 days after differentiation and real-time PCR was performed. Results are shown as fold change relative to the SEN group. Results were obtained using CM from 5 strains of human primary cells from different subjects and expressed as mean ± s.e.m. *p<0.05. Two-tailed Student's t tests were used to determine statistical significance.

The following source data is available for figure 3:

**Source data 1.** Inhibition of activin A alleviates the impairment of adipogenesis induced by senescent progenitors.

higher in he INK-ATTAC[+/-] than WT mice (**Figure 4f**), indicative of improved adipogenesis. Lipin-1, whose expression in fat tissue is positively associated with adipose tissue function (**Nadra et al., 2012**) and insulin sensitivity (**Donkor et al., 2008**) , was also increased in the INK-ATTAC[+/-] mice (**Figure 4f**). The senescence markers, *IL6, p16[Ink4a]*, and *p21[Cip1]* (**Figure 4f**) as well as SABG[+] cells (**Figure 4d** and **Figure 4—figure supplement 1**), were reduced in fat tissue of AP20187-treated INK-ATTAC[+/-] mice. These results suggest that senescent cells are a cause of age-related adipose tissue loss and dysfunction in older mice.

## JAK inhibition reduces activin A production in senescent progenitors and partially rescues adipogenesis

We recently reported that JAK inhibition suppresses SASP factors, including IL6 and TNFα, in senescent fat progenitors (**Xu et al., 2015**). We also previously observed that direct addition of recombinant activin A to cultured human fat progenitors impedes adipogenesis (**Zaragosi et al., 2010**). Here, we found that JAK inhibition reduces activin A at both the transcript (**Figure 5a**) and secreted protein levels (**Figure 5b**) in senescent fat progenitors. We therefore tested whether JAK inhibition alleviates impaired adipogenesis related to senescence. CM prepared from senescent progenitors exposed to JAK inhibitor caused less inhibition of adipogenesis in non-senescent target progenitors than CM prepared from senescent cells exposed to vehicle (**Figure 5c,d**). Since JAK inhibitor was present in the CM, we examined whether the improvement of adipogenesis in the target non-

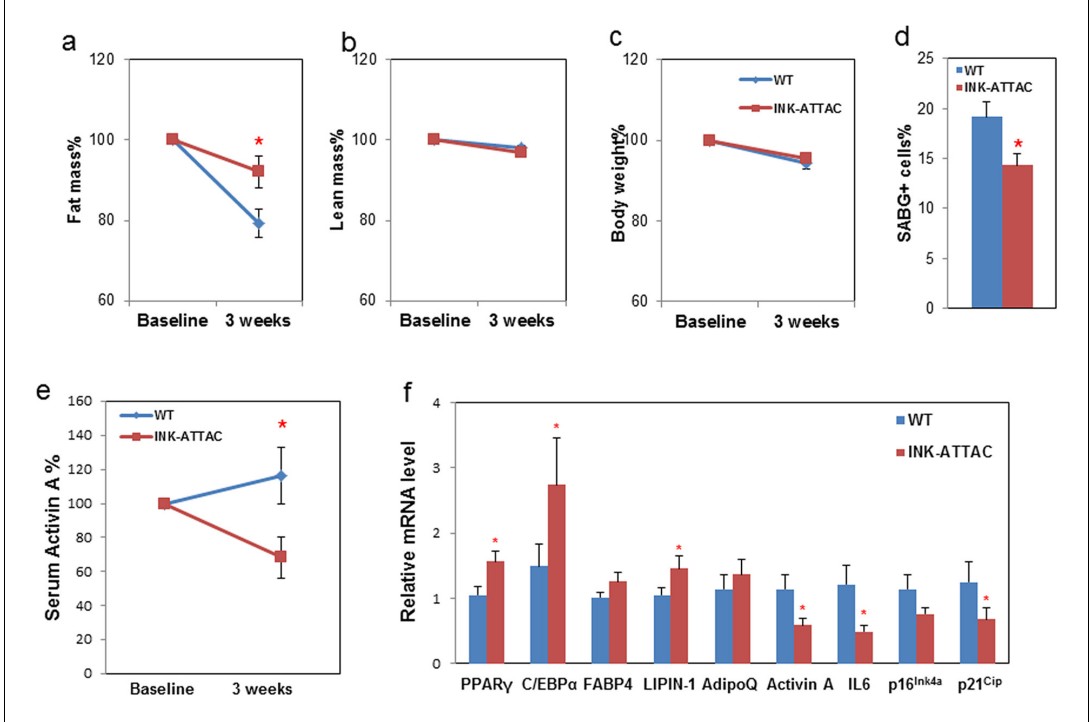

**Figure 4.** Genetic clearance of senescent cells blunts fat loss and increases adipogenic markers in fat of 18-month-old mice. Eighteen-month-old wild-type and INK-ATTAC$^{+/-}$ mice were treated with AP20187 for 3 weeks (10mg/kg, three consecutive days with 14 days rest between treatments; total 6 treatments). Fat mass (a) and lean mass (b) were measured by MRI along with body weight (c) before and after treatment. The percent changes relative to baseline are shown. Results (N=8) are expressed as mean ± s.e.m. *p<0.05 for comparison between WT and INK-ATTAC$^{+/-}$ at 3 weeks. (d) SABG$^+$ cells were counted in WAT and their percentages as a function of total cells (N=7) are expressed as mean ± s.e.m. *p<0.05. (e) Activin A protein in plasma was measured before and after treatment. The percent changes relative to baseline are shown. Results (N=8) are expressed as mean ± s.e.m. *p<0.05 for comparison between WT and INK-ATTAC$^{+/-}$ at 3 weeks. (f) RNA from white adipose tissue (WAT) was collected and real-time PCR was performed. Results (N=8) are expressed as mean ± s.e.m. *p<0.05. Two-tailed Student's t tests were used to determine statistical significance.

The following source data and figure supplements are available for figure 4:

**Source data 1.** Genetic clearance of senescent cells blunts fat loss and increases adipogenic markers in fat of 18-month-old mice.

**Figure supplement 1.** Genetic clearance of senescent cells reduced SABG$^+$ cells in adipose tissue.

**Figure supplement 2.** Senescent cell clearance blunts fat loss in 18-month INK-ATTAC$^{+/-}$ mice.

senescent cells was due to the effect of JAK inhibitor on the senescent source cells or if JAK inhibitor had direct effects on the target cells. Addition of JAK inhibitor directly to CM previously collected from either control or senescent cells did not affect adipogenesis in the target non-senescent cells (*Figure 5—figure supplement 1a*). This indicates that JAK inhibitor alleviated impaired adipogenesis mainly by acting on the senescent source fat progenitors, in turn altering the composition of the CM, rather than having direct effects on the target cells. Moreover, JAK inhibition improved adipogenesis in cultures of fat progenitors isolated from aged rats, which contain senescent cells, but not in cultured progenitors isolated from young rats (*Figure 5—figure supplement 1b and c*).

## JAK inhibition enhances adipogenesis and prevents fat loss in old mice

To test effects of JAK inhibition in vivo, we treated 22-24 month-old C57BL/6 male mice with ruxolitinib (INCB), a selective JAK1/2 inhibitor approved by the FDA, or vehicle (DMSO) for 2 months. Vehicle-treated mice progressively lost fat over two months, while JAK inhibitor administration prevented this age-related fat loss (*Figure 6a,d*). The lean mass of both groups remained unchanged (*Figure 6b,e*). The body weights of the vehicle-treated compared to the INCB-treated mice was not

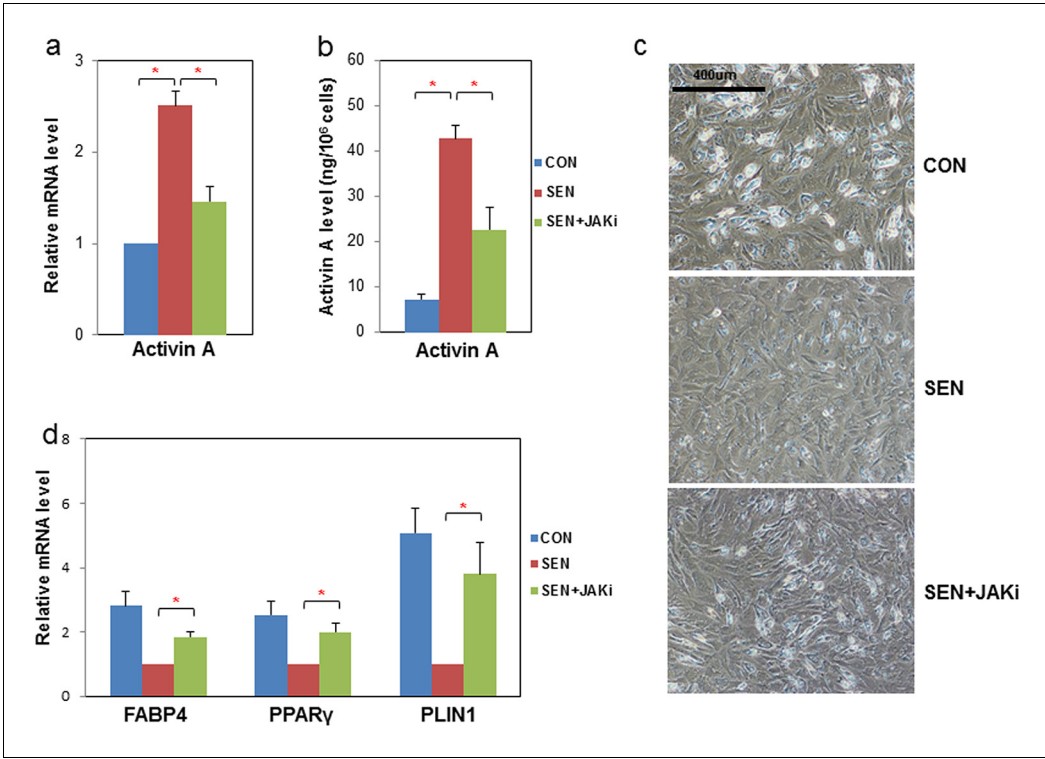

**Figure 5.** JAK inhibition suppresses activin A production by senescent fat progenitors and partially rescues adipogenesis. Senescent human progenitors were treated with DMSO (SEN) or 0.6 μM JAK inhibitor 1 (SEN+JAKi) for 72 hours. (a) RNA was collected from control (CON), SEN, and SEN+JAKi progenitors and real-time PCR was performed. Results (N=7) are expressed as mean ± s.e.m. *p<0.05. (b) CM was collected and activin A protein was assayed by ELISA. Results (N=6) are expressed as mean ± s.e.m. *p<0.05. (c) Representative images are shown of differentiating cells at day 10. (d) RNA was collected 10 days after initiation of differentiation and real-time PCR was performed. Results are shown as fold change relative to the SEN group. Results were obtained using CM from 7 strains of human primary progenitors from different subjects and expressed as mean ± s.e.m. *p<0.05. Two-tailed Student's t tests were used to determine statistical significance.

The following source data and figure supplement are available for figure 5:

**Source data 1.** JAK inhibition suppresses activin A production by senescent fat progenitors and partially rescues adipogenesis.

**Figure supplement 1.** Impaired adipogenesis due to effects of senescent cells is partially rescued by JAK inhibition.

---

significantly different (*Figure 6c,f*). This was consistent in two independent cohorts of mice using the same treatment regimen. Inguinal, subscapular, and brown fat mass were reduced in the vehicle-treated group, but were preserved in INCB-treated mice (*Figure 7a*). The same INCB treatment only exhibited a non-significant trend to alter fat mass in young (8-month-old) mice (*Figure 6—figure supplement 2a*).

We next examined the mechanism of fat mass preservation due to JAK inhibition. JAK inhibition increased adipose tissue transcript levels of the adipogenesis markers, *PPARγ, C/EBPα, FABP4, and adipo-Q*, as well as *GPAT4* (glycerol-3-phosphate acyltransferase isoform-4, a TG synthesis marker) (*Figure 7b*), suggesting that JAK inhibition may act by enhancing adipogenesis and increasing TG storage in fat in aged mice. *Lipin-1* was also increased in fat tissue from JAK inhibitor-treated mice (*Figure 7b*). Activin A increased with aging in both fat tissues (*Figure 6—figure supplement 2c*) and the circulation (*Figure 7c*). JAK inhibition suppressed *activin A* in both whole fat (*Figure 7b*) and progenitors isolated from fat tissue (*Figure 7—figure supplement 1*), as well as circulating activin A (*Figure 7c*). Notably, JAK inhibition did not reduce *activin A* expression or improve adipogenesis in

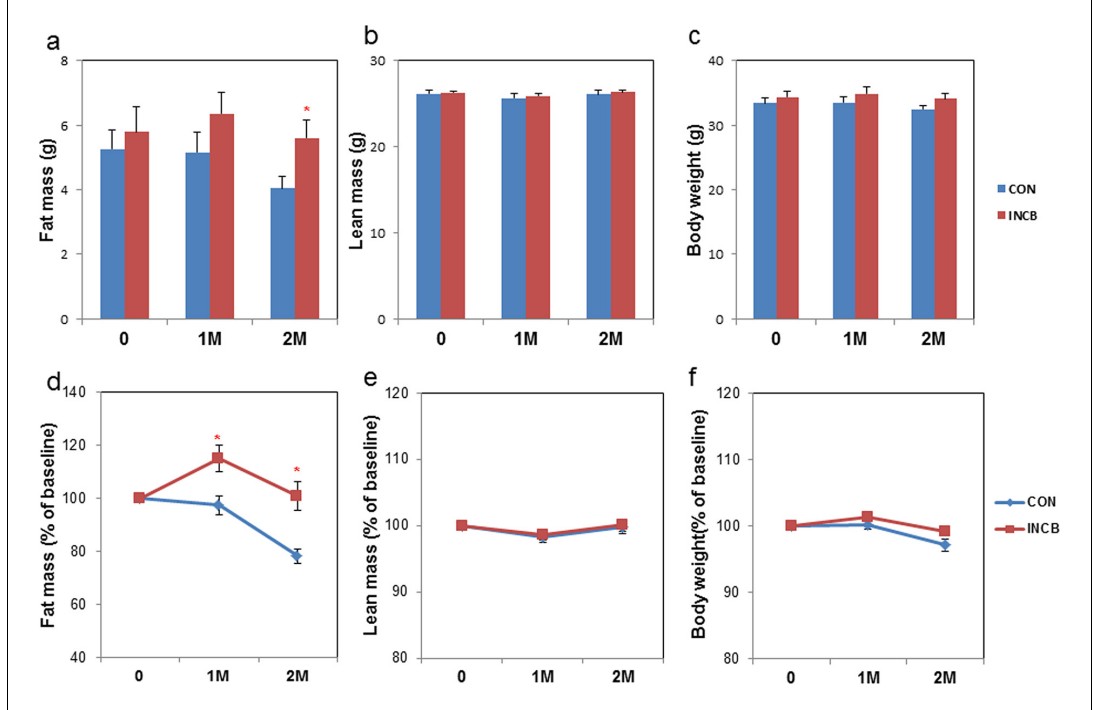

**Figure 6.** JAK inhibition reduces age-related fat loss in mice. Twenty-two-month old male mice were treated with vehicle (CON) or ruxolitinib (INCB) for 8 weeks. Fat mass (a) and lean mass (b) were measured by MRI along with body weight (c) before treatment, as well as 1 month and 2 months after treatment. The percent changes relative to baseline are shown for fat mass (d), lean mass (e), and body weight (f). Results (N=9) are expressed as mean ± s.e.m. *p<0.05. Two-tailed Student's t tests were used to determine statistical significance.

The following source data and figure supplements are available for figure 6:

**Source data 1.** JAK inhibition reduces age-related fat loss in mice.

**Figure supplement 1.** JAK inhibition did not affect metabolic rate or food intake in aged mice.

**Figure supplement 2.** JAK inhibition had less impact on body composition and adipogenesis in 8-month old mice compared to 22-month old mice.

fat tissue of younger (8-month old) mice (*Figure 6—figure supplement 2b*). We also examined other potential causes of preservation of fat mass by JAK inhibition. Administering JAK inhibitor did not change metabolic rate or food intake in aged mice (*Figure 6—figure supplement 1*), and was previously found by us to actually increase activity of old mice (*Xu et al., 2015*). Expression of two lipolytic enzymes, adipose triglyceride lipase (*ATGL*) and hormone-sensitive lipase (*HSL*), was induced in fat tissue by JAK inhibition (*Figure 7b*). This suggests that the fat maintenance we observed was not due to increased food intake, decreased energy expenditure, or decreased lipolysis. Next, we examined whether increased adipogenic capacity was associated with suppressed FA spillover and ectopic lipid accumulation. Plasma free fatty acid (FFA) levels were reduced by JAK inhibitor (*Figure 7e*), while TG was not different (*Figure 7d*). In addition, JAK inhibitor decreased both liver weight (*Figure 7a*) and hepatic TG (*Figure 7f,g*) in old mice.

## JAK inhibition enhances metabolic function in old mice

Lipotoxicity and decreased adipogenic capacity are associated with insulin resistance (*Wang et al., 2008*; *Unger and Scherer, 2010*; *Gustafson et al., 2015*; *Tchkonia et al., 2010*). We investigated whether JAK inhibitor administration enhanced insulin sensitivity in aged mice. By conducting glucose and insulin tolerance tests, we found that insulin sensitivity was impaired in 22-month- compared to 8-month-old mice (*Figure 8e*). JAK inhibitor improved glucose homeostasis (*Figure 8a,b*) and insulin sensitivity (*Figure 8d,e*) in 22-month-old mice, while it had little effect in young mice

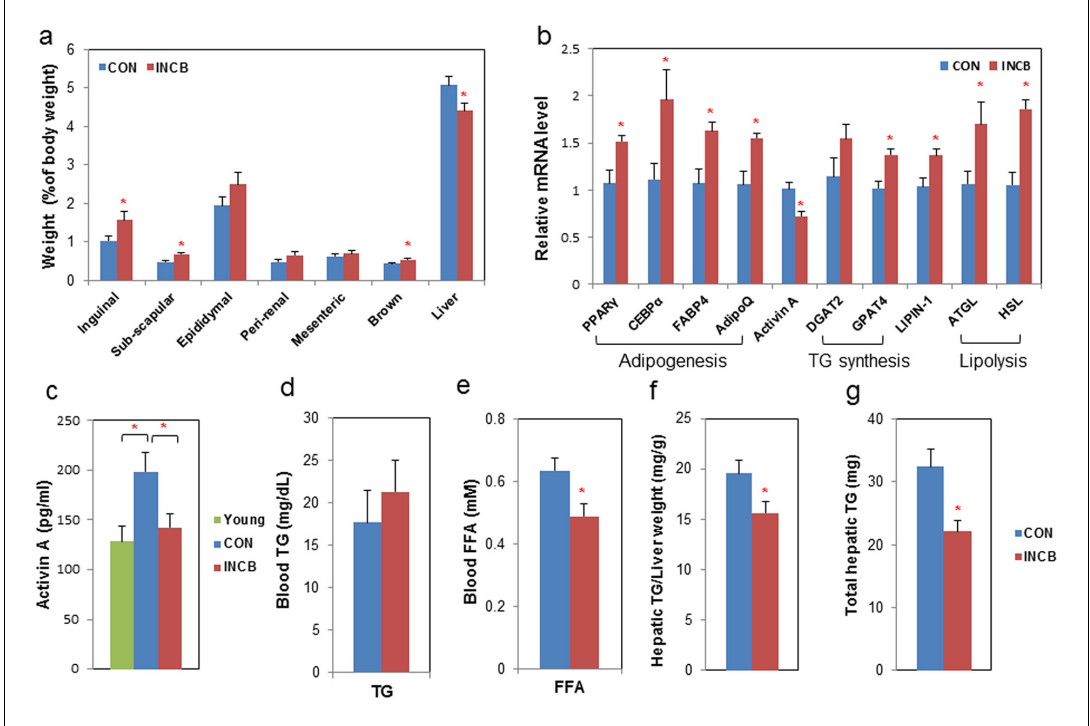

**Figure 7.** JAK inhibition increases adipogenic markers in adipose tissue and decreases circulating free fatty acids in aged mice. Twenty-two-month old male mice were treated with vehicle (CON) or ruxolitinib (INCB) for 8 weeks. (a) Weights of different fat depots and liver are shown as percent of whole body weight. Results (N=9) are expressed as mean ± s.e.m. *p<0.05. (b) RNA from WAT was isolated and real-time PCR was performed. Results (N=8) are expressed as mean ± s.e.m. *p<0.05. (c) Plasma activin A protein levels were assayed by ELISA in parallel from 8 six-month-old male mice (Young). Results (N=15 for CON and INCB, N=8 for Young) are expressed as mean ± s.e.m. *p<0.05. Plasma TG (d) and FA (e) levels are expressed as mean ± s.e.m. (N=8). *p<0.05. (f) Hepatic TG/protein levels are expressed as mean ± s.e.m. (N=11). (g) Total hepatic TG levels are expressed as mean ± s.e.m. (N=11). Two-tailed Student's t tests were used to determine statistical significance.
The following source data and figure supplement are available for figure 7:

**Source data 1.** JAK inhibition increases adipogenic markers in adipose tissue and decreases circulating free fatty acids in aged mice.
**Figure supplement 1.** JAK inhibition in aged mice suppressed activin A expression in primary fat progenitors.

(*Figure 8—figure supplement 1*). Glucose-stimulated insulin secretion capacity was not altered by JAK inhibition in 22-month-old mice (*Figure 8c*), suggesting that pancreatic islet function might not be affected. Fasting glucose was also unchanged with JAK inhibitor treatment (*Figure 8e*). To test whether insulin sensitivity in fat tissue of aged mice was improved by JAK inhibitor, we performed an *ex vivo* insulin challenge test and found that fat tissue isolated from the JAK inhibitor-treated group exhibited more robust induction of p-AKT in response to insulin compared to the control group (*Figure 8g,h*). Therefore, it appears that improved fat tissue function through JAK inhibition possibly contributed to enhanced insulin sensitivity in aged mice.

## Discussion

Adipose tissue is a key metabolic organ, dysfunction of which can be linked to metabolic disease, particularly type 2 diabetes (*Gustafson et al., 2015*). Adipose tissue function declines with age (*Tchkonia et al., 2010*), likely contributing to increased prevalence of metabolic disorders with aging. Thus, potential pharmacotherapies that alleviate age-related adipose tissue dysfunction may lead to important clinical benefit. Previously, we found that senescent cells contribute to age-related adipose tissue dysfunction in a progeroid mouse model (*Baker et al., 2011*). Here, we used progenitor cells isolated from human adipose tissue to demonstrate that senescent cells directly inhibit

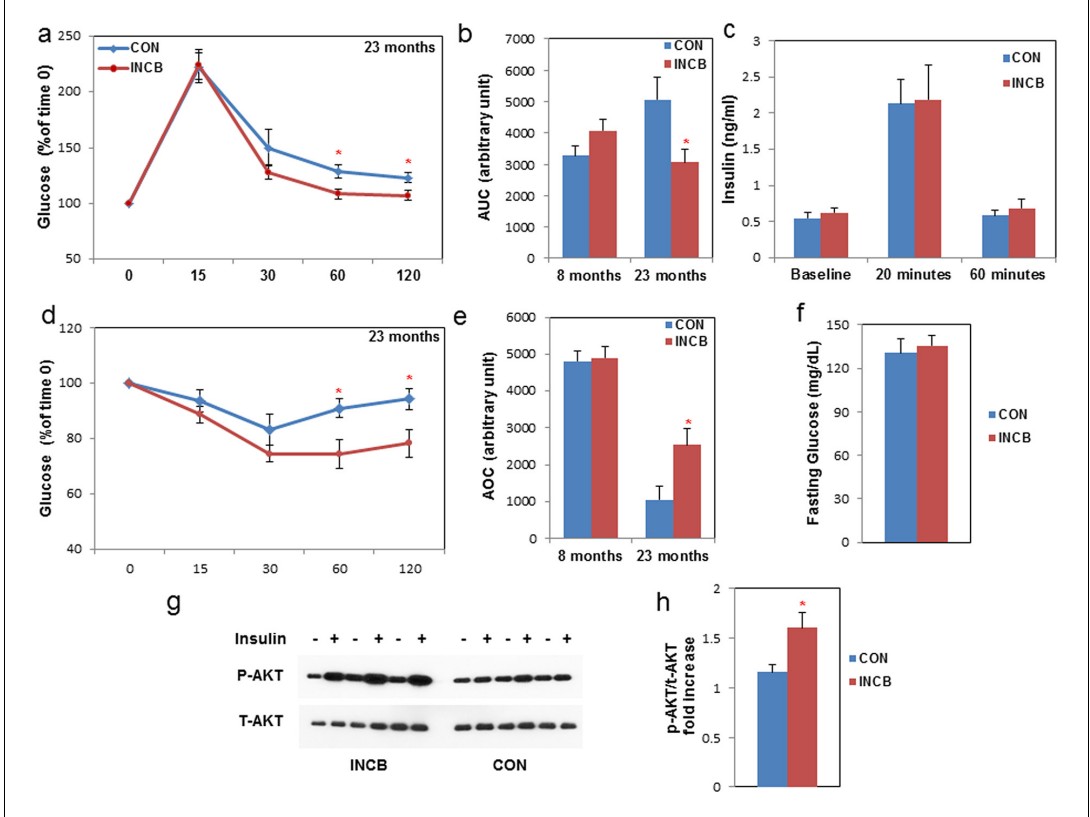

**Figure 8.** JAK inhibition increases insulin sensitivity in aged mice. Seven-month old and twenty-two-month old male mice were treated with vehicle (CON) or ruxolitinib (INCB) daily. An oral glucose tolerance test was performed after 5 weeks of treatment. (**a**) Glucose level was monitored over 120 minutes for 22-month old mice (the results for 7-month old mice are shown in *Figure 8—figure supplement 1*) and (**b**) the area under the curve (AUC) was calculated. Results (N=6 for CON and INCB groups of 8-month-old mice, N=9 for CON and INCB groups of 22-month-old mice) are expressed as mean ± s.e.m. *p<0.05. (**c**) Plasma insulin was measured at baseline, 20 minutes, and 60 minutes after oral glucose gavage. Results (N=9) are expressed as mean ± s.e.m. *p<0.05. An insulin tolerance test was performed after 6 weeks of the treatment. (**d**) Glucose was monitored over 120 minutes for 22-month old mice (the results for 7-month old mice are shown in *Figure 8—figure supplement 1*) and (**e**) area over curve (AOC) was calculated. Results (N=9) are expressed as mean ± s.e.m. *p<0.05. (**f**) Fasting glucose levels (N=9) are expressed as mean ± s.e.m. *p<0.05. (**g**) WAT tissue was collected and cultured in CM with or without 5 nM insulin for 5 minutes at 37°C and tissue lysates were then prepared. p-AKT (Ser473) and total AKT protein abundance were assayed. Representative images are shown. (**h**) These signals were quantified by densitometry using ImageJ. The ratios of p-AKT/total AKT are expressed as mean ± s.e.m. N=6. *p<0.05. Two-tailed Student's t tests were used to determine statistical significance.

The following source data and figure supplement are available for figure 8:

**Source data 1.** JAK inhibition increases insulin sensitivity in aged mice.

**Figure supplement 1.** JAK inhibition had less impact on glucose tolerance and insulin sensitivity in 8-month old mice compared to 22-month old mice.

adipogenesis of non-senescent human fat progenitors. One mechanism by which senescent cells exert this inhibitory effect is through secretion of activin A, a protein that we previously showed inhibits adipogenesis (*Zaragosi et al., 2010*). We tested the role of senescent cells in naturally-aged INK-ATTAC mice and confirmed that senescent cells play a causal role in age-related fat dysfunction in vivo. Moreover, we found that JAK inhibitor reduced activin A secretion both in vitro and in vivo. Two months of JAK inhibitor administration preserved adipose tissue function and restored insulin sensitivity in 22-month-old mice. Our study provides proof-of-concept evidence that senescent cells play an important role in age-related adipose tissue loss and dysfunction. It also suggests that inhibiting the JAK signaling pathway or selectively eliminating senescent cells hold promise as avenues to prevent or treat age-related metabolic dysfunction.

The JAK/STAT pathway plays an important role in adipose tissue development and function (*Richard and Stephens, 2014*). Previously, we found that JAK inhibitor treatment inhibited production of SASP factors and improved physical function in aged mice (*Xu et al., 2015*). Here, we show that JAK1/2 inhibition has metabolic benefits in aged mice. JAK inhibitor treatment suppressed activin A production by senescent cells in vitro and in fat progenitors, fat tissue, and the circulation in aged mice. These observations are concordant with improved adipogenesis and reduced activin A in fat tissue after genetic clearance of senescent cells from 18-month-old INK-ATTAC$^{+/-}$ mice. One possible mechanism for improved adipogenesis and fat tissue function by JAK inhibitor treatment is reduction of activin A production by senescent cells. Other SASP components (i.e. IL-6, TNFα, and IFNγ) may also contribute to impairment of adipose tissue function in aged animals.

It is possible that mechanisms other than those directly affecting senescent cells contributed to the improved adipogenesis we found. Also, senescent cells of many types and in multiple tissues, not only in fat, are likely affected by systemic administration of JAK1/2 inhibitors to mice or AP20187 to INK-ATTAC animals. Very likely, these systemic effects of our interventions contributed to alleviating metabolic dysfunction. To partially address this, rather than conducting an epistasis experiment (treating senescent cell-depleted INK-ATTAC mice with JAK inhibitor to check for off-target effects), we compared effects of JAK inhibitor treatment in old to young mice, since the latter have fewer senescent cells, like AP20187-treated INK-ATTAC mice. We feel this experiment achieves essentially the same goals as would an epistasis experiment, and arguably may even have certain advantages: potential off-target effects of AP20187 are avoided and senescent cells are very few in young mice, unlike older AP20187-treated INK-ATTAC mice, in which more than 50% of senescent cells can remain after treatment with AP20187 (*Figure 4—figure supplement 1*). The lack of substantial effects of JAK inhibitor treatment on adipogenesis, fat depot weights, and insulin sensitivity in young animals, but strong effects in old animals with higher senescent cell burden and activin A, coupled with parallel effects between JAK inhibitor treatment in wild type mice to those of genetic clearance with AP20187 in INK-ATTAC mice, suggest that effects of JAK inhibitors on senescent cells may contribute to improved metabolic function in older mice.

Our findings are consistent with the speculation that impaired adipogenesis leads to ectopic lipid accumulation and insulin resistance (*Gustafson et al., 2015*). We found that JAK pathway inhibition led to maintained fat mass and enhanced metabolic function in tandem with improved adipogenic capacity in aged mice. Expression of *PPARγ* and *C/EBPα*, both of which are essential for insulin sensitivity (*Wu et al., 1999*; *El-Jack et al., 1999*) , increased in adipose tissue of JAK inhibitor-treated mice. These changes were accompanied by reduced circulating FFAs and hepatic lipid accumulation, two important manifestations of lipotoxicity associated with insulin resistance (*Slawik and Vidal-Puig, 2006*; *Boden, 2011*). Indeed, JAK inhibition improved insulin sensitivity in these mice. In addition to improved adipogenesis, JAK inhibition reduces systemic inflammation (including reducing circulating IL6) in aged mice (*Xu et al., 2015*) and promotes 'browning' of adipose tissue (*Moisan et al., 2015*) , both of which are known to affect adipogenesis and insulin sensitivity. These mechanisms might also contribute to improved insulin sensitivity in aged mice in addition to reduced activin A level. Importantly, JAK inhibition improved adipogenesis and insulin sensitivity in aged mice but did so much less in younger mice, suggesting that the JAK pathway participates in age- or senescence-related pathogenesis of adipose tissue dysfunction. Furthermore, the effects of JAK inhibitors seem to be similar in mouse, rat, and human models. This is consistent with the speculation that the most fundamental aging mechanisms are conserved across mammalian species.

Activin A is a member of the transforming growth factor superfamily and is involved in a variety of biological events (*Xia and Schneyer, 2009*). Activin A has widespread effects on multiple types of progenitors (*Zaragosi et al., 2010*; *Bowser et al., 2013*) both directly and through interaction with the closely related growth and differentiation factors (GDFs), which share receptor and signaling mechanisms with activin A (*Sako et al., 2010*). Our results suggest that circulating activin A levels could be a bio-marker of senescent cell burden since: 1) circulating levels of activin A increase with aging, consistent with the increase in senescent cell abundance with aging, 2) senescent cells secrete activin A, 3) genetic clearance of senescent cells from 18-month-old INK-ATTAC$^{+/-}$ mice reduced circulating activin A, and 4) JAK inhibition suppresses activin A production in senescent cells in vitro and in aged mice in vivo. It is important to note that a variety of cell types can regulate activin A production, including macrophages (*Zaragosi et al., 2010*). It is possible that these cell types contribute to increased activin A levels with aging. It will be valuable to study the effect of specific inhibition of

activin A during aging. However, most activin A-blocking agents such as follistatin also inhibit myostatin due to structural similarity to activin A (*Cash et al., 2009*; *Nakatani et al., 2008*). These agents would therefore be anticipated to alter both muscle and fat mass though additional mechanisms that may be independent of activin A (*Lee, 2004*; *McPherron and Lee, 2002*).

JAK inhibitor treatment did not alter lean mass in aged mice (*Figure 6b,e*). Thus, JAK inhibition might be superior to current activin A-blocking agents for alleviation of age-related adipose tissue dysfunction. Ruxolitinib, the JAK1/2 inhibitor we used in vivo, is approved by FDA for treating myelofibrosis (*Verstovsek et al., 2010*; *Harrison et al., 2012*; *Verstovsek et al., 2012*). Although it has side-effects in human subjects with myelofibrosis including anemia and thrombocytopenia (*Verstovsek et al., 2010*; *Verstovsek et al., 2012*) , we and others found that ruxolitinib has minimal effects on peripheral blood cell populations in both young (*Quintás-Cardama et al., 2010*) and old mice (*Xu et al., 2015*). Considerable work remains to be done to assess potential side-effects from JAK1/2 inhibitors, especially in older subjects. We stress this needs to be done before contemplating their use for age-related dysfunction in clinical practice.

We observed an unusually rapid loss of fat from 18-month old INK-ATTAC$^{+/-}$ mice within 3 weeks. This fat loss could be related to the need to administer AP20187 by intraperitoneal (ip) injection, despite our making every effort to reduce this effect. Both WT and INK-ATTAC$^{+/-}$ mice were injected ip with AP20187 for three consecutive days, with 14 days between treatments. Thus, both groups received 6 ip injections within 3 weeks, the stress from which might have accelerated fat loss. Due to limited numbers of naturally aged INK-ATTAC$^{+/-}$ mice, we selected the most closely matched control group to detect an effect of clearing senescent cells on activin A and adipogenesis. The strategy of treating both the WT and INK-ATTAC$^{+/-}$ littermates with AP20187 had the advantages that both the treated and control groups received ip injection of the same drug in parallel. We used 18-month-old INK-ATTAC$^{+/-}$ mice because we have previously observed that 18 month old mice already have a detectable increase in senescent cell burden in their adipose tissue (*Stout et al., 2014*). In addition, we decided to focus on the acute effect of clearance of senescent cells from INK-ATTAC$^{+/-}$ mice on adipogenic transcription factor expression, which can precede other changes. Therefore, we decided to treat these INK-ATTAC$^{+/-}$ mice for 3 weeks. Intermittent clearance of senescent cells with AP20187 was used based on our recent finding that senolytics are effective when administered intermittently, likely because senescent cells do not divide and may be slow to re-accumulate once cleared in the absence of a strong continuing insult (*Zhu et al., 2015*). Furthermore, AP20187 has to be administered i.p., precluding daily administration. On the other hand, we showed that JAK inhibitors, which blunt the SASP and can be administered orally, need to be continuously present to inhibit the SASP (*Xu et al., 2015*).

In summary, we demonstrated a likely causal role for senescent cells in age-related fat dysfunction and discovered a novel mechanism through which senescent cells can directly impair healthy fat progenitor function. Pharmacologic inhibition of the JAK pathway reduced activin A production in vitro and in vivo, alleviated age-related adipose tissue dysfunction, and improved insulin sensitivity in aged mice. Albeit speculative, our findings are consistent with the general hypothesis that senescent cells might exert profound effects on tissue and organismal function by affecting normal progenitors or stem cells through production of TGFβ family members, such as activin A, and potentially other types of factors secreted by senescent cells. Our work suggests that targeting senescent cells or their products could be a promising avenue for delaying, preventing, alleviating, or treating age-related stem cell, progenitor, and adipose tissue dysfunction and metabolic disease.

## Materials and methods

### Cell culture and reagents

Primary human fat progenitors were isolated from subcutaneous fat collected from healthy, lean (BMI 26.6 ± 0.9 kg/m$^2$) kidney donors aged 39 ± 3.3 years as previously described (*Tchkonia et al., 2005*). The protocol (10-005236) was approved by the Mayo Clinic Foundation Institutional Review Board for Human Research. Informed consent and consent to publish was obtained from all human subjects. Rat fat progenitors were isolated from 3- and 30-month-old Brown Norway rats (purchased from Harlan Sprague Dawley) as previously described (*Tchkonia et al., 2007*). All rat and mouse experimental procedures (A21013, A37715, and A16315) were approved by the Institutional Animal

Care and Use Committee (IACUC) at Mayo Clinic. Human fat cell progenitors were subjected to 10 Gy of cesium radiation to induce senescence as described previously (*Xu et al., 2015*). Human fat cell progenitors were also treated with 0.2 µM doxorubicin for 24 hours to induce senescence. Senescence was induced by irradiation unless otherwise indicated. For co-culture experiments, primary progenitors were stained with CellTracker CM-DiI dye (Thermo Fisher Scientific, Waltham, MA, USA) according to the manufacturer's instructions. These cells were then seeded into wells containing either non-senescent control or senescent progenitors. The mixtures of cells were differentiated for 15 days. Differentiation of progenitors was assessed by observers who were not aware of which treatments the cultures had been exposed to. Cells with multiple doubly-refractile lipid inclusions visible by low power phase contrast microscopy were considered to be differentiated (*Karagiannides et al., 2006*).

JAK inhibitor 1 (CAS 457081-03-7) was purchased from EMD Millipore (Billerica, MA, USA). Ruxolitinib (INCB18424, CAS 941678-49-5) was purchased from ChemieTek (Indianapolis, IN, USA). Amicon Ultra centrifugal filters were purchased from EMD Millipore. Activin A ELISA kits (catalog number: DAC00B) and activin A neutralizing antibody (catalog number: MAB3381) were purchased from R&D Systems (Minneapolis, MN, USA). SB 431542 was purchased from Cayman Chemical (Ann Arbor, MI, USA).

## Conditioned medium collection

Cells were washed with PBS 3 times and cultured in medium to be conditioned (CM) containing 1 mM sodium pyruvate, 2 mM glutamine, MEM vitamins, MEM non-essential amino acids, and antibiotic (Thermo Fisher Scientific) for 24 hours. For JAK inhibitor treatment, cells were treated with 0.6µM JAK inhibitor or DMSO for 48 hours in regular medium, washed with PBS 3 times, and then exposed to CM containing JAK inhibitor or DMSO for another 24 hours.

## Fat progenitor differentiation

For differentiation, confluent human primary progenitors were treated with differentiation medium (DM) containing DMEM/F12, 15 nM HEPES, 15 mM NaHCO$_3$, 2 mM glutamine, 10 mg/L transferrin, 33 µM biotin, 0.5 µM insulin, 17 µM pantothenate, 0.1 µM dexamethasone, 2 nM triiodo-L-thyronine (T3), 540 µM 3-isobutyl-1-methylxanthine (IBMX), 1µM ciglitazone, 1 mg/ml fetuin, and penicillin/streptomycin for 15 days unless indicated otherwise. For conditioned medium experiments, 2x-DM was prepared by doubling the concentration of key differentiation ingredients (20 mg/L transferrin, 66µM biotin, 1 µM insulin, 34 µM pantothenate, 0.2 µM dexamethasone, 4 nM T3, 1080 µM IBMX, 2 µM ciglitazone, and 2mg/ml fetuin). Pooled cells isolated from several human subjects were then differentiated with CM mixed with 2x-DM at a 1:1 ratio for 15 days unless indicated otherwise. The media were changed every 2 days. To induce differentiation of rat cells, confluent fat progenitors were exposed to DM containing 5 µg/ml insulin, 10 µg/ml transferrin, and 0.2 nM triiodothyronine in DMEM/F-12 for 48 hours. DMEM/F12 and glutamine were purchased from Thermo Fisher Scientific. All other reagents were purchased from Sigma-Aldrich (St. Louis, MO, USA).

## Real-Time PCR

Trizol (Thermo Fisher Scientific) was used to extract RNA from tissues or cells. M-MLV Reverse Transcriptase kit (Thermo Fisher Scientific) was used for reverse transcription. Real-time PCR was performed using TaqMan fast advanced master mix. All reagents including probes and primers were purchased from Thermo Fisher Scientific. TATA-binding protein (TBP) was used as an internal control.

## Western blotting

Cells or tissues were homogenized in cell lysis buffer (Cell Signaling, Danvers, MA, USA) with protease inhibitors (Sigma-Aldrich). Coumassie Plus reagents (Pierce, Rockford, IL, USA) were used to determine total protein content. Proteins were loaded on SDS-PAGE gels and transferred to immuno-blot PVDF membranes (Biorad, Hercules, CA, USA). SuperSignal West Pico Chemiluminescent Substrate (Pierce) was used to develop signals. p-AKT (#4060) and total-AKT (#4691) antibodies were purchased from Cell Signaling.

## Comprehensive laboratory animal monitoring system and SABG activity assay

Metabolic rate and food intake were measured using a Comprehensive Laboratory Animal Monitoring System (CLAMS) as previously described (*Xu et al., 2015*). Adipose tissue cellular SABG was assayed as previously described (*Xu et al., 2015*). SABG$^+$ cells were quantified by observers who were not aware of which treatments cultures had been exposed to.

## Mice and drug treatments

Experimental procedures (A21013, A37715, and A16315) were approved by the IACUC at Mayo Clinic. Twenty two-month-old C57BL/6 male mice were obtained from the National Institute on Aging (NIA). INK-ATTAC$^{+/-}$ transgenic mice were generated and genotyped as previously described (*Baker et al., 2011*). Briefly, JLK and TT conceived the idea of clearing senescent cells to test if this improves healthspan and devised the experimental strategy of making transgenic mice with a senescence-activated promoter driving the ATTAC drug-inducible suicide gene and GFP to selectively eliminate and identify senescent cells at any time postnatally. The INK-ATTAC mice were produced and phenotyped at Mayo Clinic through a collaboration among the Kirkland, J. van Deursen, and N. LeBrasseur labs. They were bred onto a C57BL/6 background in the JVD lab. KOJ in the Kirkland lab then bred them onto a 50:50 BALB/c:C57BL/6 background, genotyped mice to select INK-ATTAC heterozygotes, and aged them to 18 months. Controls for the INK-ATTAC experiments were INK-ATTAC-null; 50:50 BALB/c:C57BL/6 background mice raised in parallel. Mice were maintained under a 12 hour light and 12 hour dark cycle at 24°C with free access to food (standard mouse diet, Lab Diet 5053, St. Louis, MO, USA) and water in a pathogen-free facility. For drug treatment, ruxolitinib was dissolved in DMSO and then mixed with food. In addition to regular food, each mouse was fed a small amount of food (0.5g) containing ruxolitinib 60 mg/kg (drug/body weight) or DMSO daily. During the treatment, all mice consumed the drug-containing food completely every day. For AP20187 (10mg/kg) treatment, drug was administered by i.p. injection for three consecutive days, with 14 days between treatments. Intermittent clearance of senescent cells with AP20187 was used based on our recent finding that senolytics are effective when administered intermittently (*Zhu et al., 2015*).

## Metabolic parameter measurement

For oral glucose tolerance testing, mice were fasted for 6 hours and glucose (2 g/kg body weight) was administrated by oral gavage. For insulin tolerance testing, mice were fasted for 4 hours and insulin (0.6 unit/kg body weight) was injected intraperitoneally. Glucose was measured using a hand-held glucometer (Bayer) in blood from the tail vein. For the glucose-stimulated insulin secretion assay, mice were fasted for 6 hours and glucose (2 g/kg body weight) was administrated by oral gavage. Blood samples were collected at baseline, 20 minutes, and 60 minutes after glucose administration. Plasma insulin levels were measured by ELISA (ALPCO, Salem, NH, USA). Fat and lean mass were measured by MRI (Echo Medical Systems, Houston, TX, USA). Hepatic TG was measured as previously described (*Xu et al., 2011*). FFA and TG were measured using kits from Wako Chemicals (Richmond, VA, USA). In all studies, investigators conducting analyses of animals were not aware of which treatments animals had received.

## Statistical methods

Two-tailed Student's t tests were used to determine statistical significance. $p<0.05$ was considered significant. All values are expressed as mean ± s.e.m. No randomization was used to assign experimental groups. We determined the sample size based on our previous experiments, so no statistical power analysis was used. All replicates in this study were independent biological replicates, which came from different biological samples.

## Acknowledgements

The authors are grateful to C. Guo for helping with ruxolitinib administration, J Armstrong for administrative assistance, and M Mahlman for obtaining human fat samples. This work was supported by NIH grants AG13925 (JLK), AG041122 (JLK), AG31736 (Project 4: JLK), AG044396 (JLK),

DK50456 (JLK), AG46061 (AKP), the Connor Group, and the Glenn, Ted Nash Long Life, and Noaber Foundations (JLK). MX received a Glenn/American Federation for Aging Research Postdoctoral Fellowship for Translational Research on Aging.

## Additional information

### Competing interests

AKP, TP, NG, TT and JLK: This research has been reviewed by the Mayo Clinic Conflict of Interest Review Board and is being conducted in compliance with Mayo Clinic Conflict of Interest policies. The other author declares that no competing interests exist.

### Funding

| Funder | Author |
| --- | --- |
| National Institute on Aging | Michael D Jensen<br>Nathan K LeBrasseur<br>James L Kirkland |
| Glenn Foundation for Medical Research | Nathan K LeBrasseur<br>James L Kirkland |

The funders had no role in study design, data collection and interpretation, or the decision to submit the work for publication.

### Author contributions

MX, conceived the project and designed the experiments, performed animal studies, performed cell culture studies; analyzed the data, wrote the manuscript, revised and approved the manuscript, Conception and design, Acquisition of data, Analysis and interpretation of data, Drafting or revising the article, Contributed unpublished essential data or reagents; AKP, performed animal studies, analyzed the data, wrote the manuscript, revised and approved the manuscript, Acquisition of data, Analysis and interpretation of data, Drafting or revising the article; HD, TAW, performed animal studies, analyzed the data, revised and approved the manuscript, Acquisition of data, Analysis and interpretation of data, Drafting or revising the article; MMW, performed cell culture studies, analyzed the data, revised and approved the manuscript, Acquisition of data, Analysis and interpretation of data, Drafting or revising the article; TP, KOJ, MBS, performed animal studies, revised and approved the manuscript, Acquisition of data, Analysis and interpretation of data, Drafting or revising the article; AS, revised and approved the manuscript, Acquisition of data, Analysis and interpretation of data, Drafting or revising the article; NG, contributed to isolation of primary human preadipocytes; revised and approved the manuscript, Acquisition of data, Analysis and interpretation of data, Drafting or revising the article; MDJ, contributed to isolation of primary human preadipocytes, contributed to manuscript preparation, revised and approved the manuscript, Acquisition of data, Analysis and interpretation of data, Drafting or revising the article, Contributed unpublished essential data or reagents; NKL, contributed to manuscript preparation, revised and approved the manuscript, Analysis and interpretation of data, Drafting or revising the article, Contributed unpublished essential data or reagents; TT, conceived the project and designed the experiments; performed animal studies, wrote the manuscript; oversaw all experimental design, data analysis, and manuscript preparation, revised and approved the manuscript, Conception and design, Acquisition of data, Analysis and interpretation of data, Drafting or revising the article; JLK, conceived the project and designed the experiments, wrote the manuscript, oversaw all experimental design, data analysis, and manuscript preparation; revised and approved the manuscript, Conception and design, Drafting or revising the article, Contributed unpublished essential data or reagents

### Ethics

Human subjects: The protocol (10-005236) was approved by the Mayo Clinic Foundation Institutional Review Board for Human Research. Informed consent and consent to publish was obtained from all human subjects.

Animal experimentation: Experimental procedures (A21013, A37715 and A16315) were approved by the IACUC at Mayo Clinic.

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
