## [Decision Letter]

Thank you for submitting your work entitled "Targeting Senescent Cells Enhances Adipogenesis and Metabolic Function in Old Age" for peer review at *eLife*. Your article has been favorably evaluated by Sean Morrison (Senior Editor) and two reviewers: Andrew Dillin, who also served as Reviewing Editor, and Amy Wagers.

The reviewers have discussed the reviews with one another and the Reviewing Editor has drafted this decision to help you prepare a revised submission.

Summary:

The manuscript by Kirkland and colleagues investigates the influence of senescent cells on adiposity in aged animals. The authors present evidence that soluble factors produced by senescent cells inhibit the differentiation of non-senescent adipogenic progenitors, and, using a transgenic system that ablates senescent cells in vivo, they show that reducing the endogenous burden of senescent cells prevents age-related loss of fat mass. The manuscript is well-written, clearly describes the approach and rationale for the work (although the INK-ATTAC system could be described in a bit more detail here, rather than simply referring to past work), and presents a logical series of experiments. The demonstration that senescent cell ablation rescues a phenotype of normal aging is a significant advance, which has been anticipated by the aging research community since the original publication of the INK-ATTAC allele and its effects in a progeroid mouse model. In the current paper, the authors also investigate the potential molecular mediators of the effects of senescent cells on adipose tissue. They implicate activin A as a critical mediator of the suppression of adipogenesis by senescent cells and suggest that its production by senescent cells requires activation of JAK signaling. Consistent with this, JAK inhibition reduces activin A production and partially rescues adipogenesis, restores fat mass, and increases insulin sensitivity in aged mice.

Essential revisions:

The authors do not fully close the loop on the proposed mechanism, as the manuscript in its current form does not fully link the effects of JAK inhibition in vivo with JAK inhibition in senescent cells. In other words, it remains possible that inhibition of JAK signaling in other cell types contributes to the observed phenotypes, a caveat that the authors do mention. Ideally, this caveat would be addressed experimentally (e.g. by treating the INK-ATAAC mice with JAKi), especially since the basic components of the mechanism were already known.

While both reviewers agree that this experiment would put this paper over the edge, we agree that the paper should be accepted if the authors temper their claims of mechanism within the text, highlighting this point (unless of course this data is in hand).

Lastly, both reviewers had issues with the weight gain studies. This point needs to be addressed (see details below).

Reviewer #1:

Following-up on a previous study, Xu and co-workers investigated the nature of the signal secreted by senescent cells to cause age-dependent lipodystrophy and decline in glucose homeostasis. In an extensive body of work the authors propose that the secreted factor is Activin A and acts via a downstream effector in adipose tissue, the JAK signaling pathway. Using a pharmacologic approach, they show that JAK signaling interferes with differentiation of non-senescent progenitor cells, and inhibits adipogenesis in vitro. The strength of this paper relies on in vivo mouse models where pharmacological inhibition of Activin-A/JAK reduces lipotoxicity and increased insulin sensitivity in aged mice. However, in order to be accepted for publication, two major concerns must be addressed:

1) Specificity of Activin A in the senescent response:

When inhibiting SASP factors, only results implicating Activin A as an adipogenesis inhibitor are presented (Figure 3). However, the text also mentions testing IL6, TNFα, IFNγ. Please provide the data in the figure. Additionally, in Figure 5, it is shown that JAK inhibits Activin A production by senescent fat progenitors. Other potential mediators of the senescent response are dismissed. These data must be presented in order to consolidate the role of Activin A in the senescent response. Later in the paper, JAK inhibition of the Activin A pathway is presented as a partial way to rescue adipogenesis and improve metabolic parameters in old mice (Figure 6 and Figure 7). This implies that other factors are also participating in the degradation of adipogenesis with age. Please speculate in the Discussion on other SASP factors that might work together with Activin A to affect lipotoxicity with age.

2) Mouse studies design:

Why are the wild-type mice loosing fat so rapidly at 18 months in just 3 weeks in Figure 4? Other studies in C57BL/6 do not report such a dramatic body weight change with age (Selan et al., Science 2009, Ortega-Molina et al., Cell Metab 2012, Riera et al., Cell 2014, etc.). It seems highly unlikely that fat loss with age occurs so dramatically in the course of 3 weeks. Also, a vehicle control for the INK-ATTAC and WT mice must be provided in addition to drug treatment.

Later in the manuscript, treatment with the JAK inhibitor is assessed in older animals (22 months) over the course of 8 weeks. The experimental procedure used here is more convincing to observe a change in adipogenesis and glucose metabolism with age than the INK-ATTAC experiment. Can the authors comment on why they used different ages, and different latencies of drug treatment in INK-ATTAC versus JAK inhibitor treatment?

Reviewer #2:

The manuscript by Kirkland and colleagues investigates the influence of senescent cells on adiposity in aged animals. The authors present evidence that soluble factors produced by senescent cells inhibit the differentiation of non-senescent adipogenic progenitors, and, using a transgenic system that ablates senescent cells in vivo, they show that reducing the endogenous burden of senescent cells prevents age-related loss of fat mass. The manuscript is well-written, clearly describes the approach and rationale for the work (although the INK-ATTAC system could be described in a bit more detail here, rather than simply referring to past work), and presents a logical series of experiments. The demonstration that senescent cell ablation rescues a phenotype of normal aging is a significant advance, which has been anticipated by the aging research community since the original publication of the INK-ATTAC allele and its effects in a progeroid mouse model. In the current paper, the authors also investigate the potential molecular mediators of the effects of senescent cells on adipose tissue. They implicate activin A as a critical mediator of the suppression of adipogenesis by senescent cells and suggest that its production by senescent cells requires activation of JAK signaling. Consistent with this, JAK inhibition reduces activin A production and partially rescues adipogenesis, restores fat mass, and increases insulin sensitivity in aged mice. These data are interesting, but suffer somewhat from the fact that the authors have reported key aspects of the proposed signaling system previously, including the demonstration that activin A is produced by senescent cells as a component of the SASP, that activin A antagonizes adipogenic differentiation and that JAK inhibition reduced SASP production and promoted physiological changes consistent with improved metabolic function. Thus, while this paper is useful in "connecting the dots", mechanistically, it can be argued that the novel data relate mostly to the senescent cell ablation studies, and in this regard, the authors do not fully close the loop on the proposed mechanism, as the manuscript in its current form does not fully link the effects of JAK inhibition in vivo with JAK inhibition in senescent cells. In other words, it remains possible that inhibition of JAK signaling in other cell types contributes to the observed phenotypes, a caveat that the authors do mention. Ideally, this caveat would be addressed experimentally (e.g. by treating the INK-ATAAC mice with JAKi), especially since the basic components of the mechanism were already known. Still, the key advance provided by the demonstration that senescent cell ablation has physiologic benefit in naturally aged animals should not be overlooked, and is a compelling aspect of the manuscript.

---

## [Author Response]

Summary:

*The manuscript by Kirkland and colleagues investigates the influence of senescent cells on adiposity in aged animals. The authors present evidence that soluble factors produced by senescent cells inhibit the differentiation of non-senescent adipogenic progenitors, and, using a transgenic system that ablates senescent cells in vivo, they show that reducing the endogenous burden of senescent cells prevents age-related loss of fat mass. The manuscript is well-written, clearly describes the approach and rationale for the work (although the INK-ATTAC system could be described in a bit more detail here, rather than simply referring to past work), and presents a logical series of experiments. The demonstration that senescent cell ablation rescues a phenotype of normal aging is a significant advance, which has been anticipated by the aging research community since the original publication of the INK-ATTAC allele and its effects in a progeroid mouse model. In the current paper, the authors also investigate the potential molecular mediators of the effects of senescent cells on adipose tissue. They implicate activin A as a critical mediator of the suppression of adipogenesis by senescent cells and suggest that its production by senescent cells requires activation of JAK signaling. Consistent with this, JAK inhibition reduces activin A production and partially rescues adipogenesis, restores fat mass, and increases insulin sensitivity in aged mice.*

We explained the INK-ATTAC model in more detail.

*Essential revisions:*

*The authors do not fully close the loop on the proposed mechanism, as the manuscript in its current form does not fully link the effects of JAK inhibition in vivo with JAK inhibition in senescent cells. In other words, it remains possible that inhibition of JAK signaling in other cell types contributes to the observed phenotypes, a caveat that the authors do mention. Ideally, this caveat would be addressed experimentally (e.g. by treating the INK-ATAAC mice with JAKi), especially since the basic components of the mechanism were already known.*

We agree that effects of JAK inhibition on senescent cells might not be the only mechanism for its in vivo effects. Rather than conducting an epistasis experiment (treating senescent cell-depleted INK-ATTAC mice with JAK inhibitor to check for off-target effects), we instead compared effects of JAK inhibitor treatment in old to young mice, since the latter, like AP20187-treated INK-ATTAC mice, have fewer senescent cells. In the young mice, treating with the JAK inhibitor did not reduce activin A expression or improve adipogenesis in their fat tissue. Insulin sensitivity was not altered by JAK inhibition in the young animals either. We believe this piece of new evidence further supports our proposed mechanism for JAK inhibition in vivo. We feel this experiment achieves essentially the same goals as the suggested epistasis experiment, and arguably may even have certain advantages: potential off-target effects of AP20187 are avoided and senescent cells are very few in young mice, unlike older AP20187-treated INK-ATTAC mice, in which around 50% of senescent cells remain after treatment with AP20187. We included these new data and discussion about these points in the revised manuscript.

We also point out in the revised Discussion that, as the editor correctly suggests, JAK inhibitors most likely also have effects on multiple cell types besides senescent cells. These effects could well contribute to our findings. However, the lack of substantial effects on adipogenesis, fat depot weights, and insulin sensitivity in young animals, but strong effects in old animals with higher senescent cell burden and activin A, coupled with parallel effects between JAK inhibitor treatment in wild type mice to those of genetic clearance with AP20187 in INK-ATTAC mice, indicate that effects of JAK inhibitors on senescent cells potentially contribute to improved metabolic function in older mice.

While both reviewers agree that this experiment would put this paper over the edge, we agree that the paper should be accepted if the authors temper their claims of mechanism within the text, highlighting this point (unless of course this data is in hand).

We tempered our claims of proposed mechanism in the Abstract, Results, and Discussion. We also included new data about the effects of JAK inhibitors in young mice that have a low burden of senescent cells.

Lastly, both reviewers had issues with the weight gain studies. This point needs to be addressed (see details below).

We addressed this (see below).

Reviewer #1:

*Following-up on a previous study, Xu and co-workers investigated the nature of the signal secreted by senescent cells to cause age-dependent lipodystrophy and decline in glucose homeostasis. In an extensive body of work the authors propose that the secreted factor is Activin A and acts via a downstream effector in adipose tissue, the JAK signaling pathway. Using a pharmacologic approach, they show that JAK signaling interferes with differentiation of non-senescent progenitor cells, and inhibits adipogenesis* in vitro*. The strength of this paper relies on* in vivo

*mouse models where pharmacological inhibition of Activin-A/JAK reduces lipotoxicity and increased insulin sensitivity in aged mice. However, in order to be accepted for publication, two major concerns must be addressed: 1) Specificity of Activin A in the senescent response:*

*When inhibiting SASP factors, only results implicating Activin A as an adipogenesis inhibitor are presented (Figure 3). However, the text also mentions testing IL6, TNFα, IFNγ. Please provide the data in the figure. Additionally, in Figure 5, it is shown that JAK inhibits Activin A production by senescent fat progenitors. Other potential mediators of the senescent response are dismissed. These data must be presented in order to consolidate the role of Activin A in the senescent response. Later in the paper, JAK inhibition of the Activin A pathway is presented as a partial way to rescue adipogenesis and improve metabolic parameters in old mice (Figure 6 and Figure 7). This implies that other factors are also participating in the degradation of adipogenesis with age. Please speculate in the Discussion on other SASP factors that might work together with Activin A to affect lipotoxicity with age.*

We provided the requested IL6, TNFα, and IFNγ data in Figure 2—figure supplement 1. In our recently published work, we showed that JAK inhibitors suppress IL6 and TNFα secretion by senescent cells (but not for IFNγ; Xu, M., et al. Proc. Natl. Acad. Sci. (USA); published ahead of print November 2, 2015, doi:10.1073/pnas.1515386112). We included this information in our revised manuscript. We also discussed the potential role of these factors in age-related metabolic dysfunction in the Discussion.

2) Mouse studies design:

*Why are the wild-type mice loosing fat so rapidly at 18 months in just 3 weeks in Figure 4? Other studies in C57BL/6 do not report such a dramatic body weight change with age (Selan et al., Science 2009, Ortega-Molina et al., Cell Metab 2012, Riera et al., Cell 2014, etc.). It seems highly unlikely that fat loss with age occurs so dramatically in the course of 3 weeks. Also, a vehicle control for the INK-ATTAC and WT mice must be provided in addition to drug treatment.*

We agree with that the rapid fat loss in our INK-ATTAC mouse model is unusual. We believe this rapid fat loss could be related to the need to administer AP20187 by intraperitoneal (ip) injection, despite our making every effort to reduce this effect. Both wild type (WT) and INK-ATTAC mice were injected ip with AP20187 for three consecutive days, with 14 days between treatments. Thus, both groups received 6 ip injections within 3 weeks, the stress from which might have accelerated fat loss.

We also agree that a vehicle control for the INK-ATTAC and WT mice would be desirable. However, due to limited numbers of naturally aged INK-ATTAC mice, we had to select the most closely matched control group to detect an effect of clearing senescent cells on activin A and adipogenesis. The strategy of treating both the WT and INK-ATTAC littermates with AP20187 had the advantages that both the treated and control groups received ip injection of the same drug in parallel and this strategy controls for any off-target effects of AP20187. We included these points in the revised Discussion.

*Later in the manuscript, treatment with the JAK inhibitor is assessed in older animals (22 months) over the course of 8 weeks. The experimental procedure used here is more convincing to observe a change in adipogenesis and glucose metabolism with age than the INK-ATTAC experiment. Can the authors comment on why they used different ages, and different latencies of drug treatment in INK-ATTAC versus JAK inhibitor treatment?*

We were able to obtain 22 month old WT mice from NIA, while we needed to generate older INK-ATTAC mice ourselves. We only had 18-month-old INK-ATTAC mice available. From previous work (Stout, M.B., et al., Aging 6:575-586, 2014), we knew that 18-month-old mice already have detectible senescent cell burden in their adipose tissue. Also, we decided to focus on the acute effect of clearance of senescent cells from INK-ATTAC mice on adipogenic transcription factor expression, which can precede other changes. Therefore, we decided to treat these INK-ATTAC mice for 3 weeks, the time at which we had first observed detectible effects of JAK inhibitors. We added these points to the revised Discussion.

We used intermittent treatment with AP20187 and continuous treatment with JAK inhibitors. Intermittent clearance of senescent cells with AP20187 was used based on our recent finding that senolytics are effective when administered periodically, likely because senescent cells do not of course divide and may be slow to re-accumulate once cleared in the absence of a strong continuing insult (Zhu, Y., et al., Aging Cell 14:644-658, 2015. PMID: 25754370). Furthermore, the AP20187 has to be administered ip, precluding daily administration. On the other hand, we showed that JAK inhibitors, which blunt the SASP and can be administered orally, need to be continuously present to inhibit the SASP (Xu, M., et al., Proc. Natl. Acad. Sci. (USA); published ahead of print November 2, 2015, doi:10.1073/pnas.1515386112). We explained our rationale for the different administration strategies in the revised Methods and Discussion.

Reviewer #2:

*The manuscript by Kirkland and colleagues investigates the influence of senescent cells on adiposity in aged animals. The authors present evidence that soluble factors produced by senescent cells inhibit the differentiation of non-senescent adipogenic progenitors, and, using a transgenic system that ablates senescent cells in vivo, they show that reducing the endogenous burden of senescent cells prevents age-related loss of fat mass. The manuscript is well-written, clearly describes the approach and rationale for the work (although the INK-ATTAC system could be described in a bit more detail here, rather than simply referring to past work), and presents a logical series of experiments. The demonstration that senescent cell ablation rescues a phenotype of normal aging is a significant advance, which has been anticipated by the aging research community since the original publication of the INK-ATTAC allele and its effects in a progeroid mouse model. In the current paper, the authors also investigate the potential molecular mediators of the effects of senescent cells on adipose tissue. They implicate activin A as a critical mediator of the suppression of adipogenesis by senescent cells and suggest that its production by senescent cells requires activation of JAK signaling. Consistent with this, JAK inhibition reduces activin A production and partially rescues adipogenesis, restores fat mass, and increases insulin sensitivity in aged mice. These data are interesting, but suffer somewhat from the fact that the authors have reported key aspects of the proposed signaling system previously, including the demonstration that activin A is produced by senescent cells as a component of the SASP, that activin A antagonizes adipogenic differentiation and that JAK inhibition reduced SASP production and promoted physiological changes consistent with improved metabolic function. Thus, while this paper is useful in "connecting the dots", mechanistically, it can be argued that the novel data relate mostly to the senescent cell ablation studies, and in this regard, the authors do not fully close the loop on the proposed mechanism, as the manuscript in its current form does not fully link the effects of JAK inhibition in vivo with JAK inhibition in senescent cells. In other words, it remains possible that inhibition of JAK signaling in other cell types contributes to the observed phenotypes, a caveat that the authors do mention. Ideally, this caveat would be addressed experimentally (e.g. by treating the INK-ATAAC mice with JAKi), especially since the basic components of the mechanism were already known. Still, the key advance provided by the demonstration that senescent cell ablation has physiologic benefit in naturally aged animals should not be overlooked, and is a compelling aspect of the manuscript.*

We described the INK-ATTAC model in more detail in the revised Introduction.

We appreciate the positive comments and the acknowledgement of importance of senescent cell clearance in naturally aged mice. In our recently published work, we showed that senescent human preadipocytes developed a SASP, which can in part be suppressed by JAK inhibitors (Xu, M., et al., Proc. Natl. Acad. Sci. (USA); published ahead of print November 2, 2015, doi:10.1073/pnas.1515386112). However, activin A was not studied in that paper. Here, we showed that senescent preadipocytes secrete more activin A than non-senescent cells. We also show that activin A secretion by senescent cells can be decreased by JAK inhibitors for the first time. We report that JAK inhibition has metabolic benefits specifically in aged, but not young, mice for the first time as far as we are aware. Restoration of adipogenesis by clearing senescent cells is shown here for the first time. In new data added to the paper, we demonstrate an age-related increase in adipose tissue activin A (Figure 6—figure supplement 2). Nevertheless, as mentioned above, we fully agree that the loop of our proposed mechanism is not completely closed. As suggested by the reviewer, we revised our manuscript accordingly.